# Clinical Reasoning and Practices in the Osteopathic Management of Visceral Disorders: A Grounded Theory Study in the Italian Context

**DOI:** 10.3390/healthcare13161995

**Published:** 2025-08-14

**Authors:** Tommaso Camonico, Francesca Lippi, Nicolò Rizzo, Alessio Barusso, Giacomo Rossettini, Jorge Hugo Villafañe, Francesco Cerritelli, Liria Papa, Jorge E. Esteves

**Affiliations:** 1Malta ICOM Educational, GZR 1075 Gzira, Malta; francesca.lippi@studenti.icomedicine.com (F.L.); nicolo.rizzo@studenti.icomedicine.com (N.R.); alessio.barusso@studenti.icomedicine.com (A.B.); liria.papa@icomedicine.com (L.P.); jesteves@uatlantica.pt (J.E.E.); 2School of Physiotherapy, University of Verona, 37129 Verona, Italy; giacomo.rossettini@gmail.com; 3Department of Physiotherapy, Faculty of Medicine, Health and Sports, Universidad Europea de Madrid, 28670 Villaviciosa de Odón, Spain; mail@villafane.it; 4Foundation COME Collaboration, Clinical-Based Human Research Department, 65126 Pescara, Italy; francesco.cerritelli@gmail.com; 5New York Institute of Technology, College of Osteopathic Medicine, Old Westbury, NY 11568, USA; 6Escola Superior de Saúde Atlântica, Fábrica da Pólvora de Barcarena, 2730-036 Barcarena, Portugal

**Keywords:** visceral osteopathy, clinical reasoning, grounded theory, qualitative study, Italian osteopathy

## Abstract

Background and Rationale: Visceral disorders, both functional and organic, significantly impact health-related quality of life and pose a challenge for healthcare systems. Osteopathy offers manual, systemic, and patient-centered approaches for their management, yet these remain controversial due to limited scientific support and methodological inconsistencies. In the evolving landscape of healthcare regulation in Europe, and particularly in Italy, exploring clinical reasoning and operational models in visceral osteopathy is essential. This study aimed to explore the beliefs, clinical reasoning, and management strategies of experienced Italian osteopaths in the treatment of visceral disorders using a Constructivist Grounded Theory approach. Methods: This qualitative study applied a Constructivist Grounded Theory approach to explore the beliefs and clinical practices of 10 experienced Italian osteopaths. Semi-structured interviews were transcribed, coded, and thematically analyzed, integrating literature comparisons to support theory generation. Results: Four core themes emerged: (1) education and professional development, (2) definition and identification of visceral disorders, (3) clinical management strategies, and (4) multidisciplinary collaboration. The findings reveal marked heterogeneity in diagnostic frameworks and treatment rationales, often driven by tradition and subjective interpretation rather than empirical evidence. Palpatory assessments were frequently prioritized over patient-reported outcomes. Conclusion: The study highlights substantial fragmentation in Italian visceral osteopathic practice, echoing challenges across Europe. Promoting a shift towards critical thinking, evidence-based models, shared terminology, and interprofessional integration is essential for contextualising osteopathic contributions to the care of individuals presenting with visceral-related problems. These findings provide insights into the fragmented clinical practices of Italian osteopaths and may contribute to shaping a more critical and evidence-informed approach within Italian osteopathic practice and professional development, which is now more relevant than ever, given the recent integration of osteopathy into the Italian higher education system.

## 1. Introduction

Visceral disorders encompass a wide array of conditions that are broadly categorized as functional or organic. Functional disorders are defined by the persistence of symptoms in the absence of detectable structural abnormalities on standard diagnostic imaging or laboratory tests. In contrast, organic disorders are characterized by identifiable pathological changes [1]. Both categories may significantly compromise quality of life, particularly when chronic visceral pain is present. According to the 2019 classification of the International Association for the Study of Pain (IASP), chronic visceral pain is among the most debilitating forms of pain and is frequently associated with chronic musculoskeletal (MSK) pain [2,3]. Additionally, visceral pain may manifest as referred pain in somatic structures, further complicating its diagnosis and management [3].

Functional visceral disorders—currently defined as “Disorders of Gut–Brain Interaction” (DGBI)—are highly prevalent, affecting approximately 40% of the global population. These conditions are linked to dysregulation in the central nervous system’s (CNS) control of visceral function and often co-occur with psychological and somatic comorbidities such as anxiety, depression, insomnia, and fibromyalgia [4,5]. On the other hand, organic disorders such as inflammatory bowel disease (IBD), organic dyspepsia, and endometriosis, although etiologically distinct, often coexist with functional disorders, presenting diagnostic and therapeutic challenges in clinical settings [6].

Current treatment strategies typically involve a multidisciplinary approach, integrating pharmacological therapy, surgical interventions, dietary modifications, probiotic supplementation, psychotherapy, and physical activity [1,7,8].

Within this clinical context, osteopathy has gained recognition as a complementary and alternative medical approach, defined by the World Health Organization (WHO) as a system employing manual techniques to assess and treat somatic dysfunction. This concept remains the subject of ongoing debate due to the lack of a clear pathophysiological foundation [9,10,11]. Osteopathy is grounded in a patient-centered philosophy aimed at supporting the body’s inherent capacity for self-regulation and healing [12]. Among the range of osteopathic techniques, visceral manipulation is commonly utilized and is included in international training benchmarks for osteopathic practice. Many osteopaths regard these techniques as fundamental to a holistic approach to care and express concern over practitioners who dismiss or lack training in this area [13].

Despite their widespread use, however, the evidence supporting the clinical efficacy of visceral techniques remains inconclusive. Systematic reviews have reported no statistically significant therapeutic benefit from visceral osteopathic interventions [14,15]. This has given rise to a diversity of professional perspectives. While a substantial number of osteopaths continue to endorse the use of visceral techniques based on clinical experience, others question their validity, citing limited empirical support and low inter-practitioner reliability. A third group acknowledges the weak evidence base but notes perceived improvements in patient outcomes. Such heterogeneity of opinion may be attributed to inconsistencies in educational standards and training pathways, particularly across different national contexts [13].

Given the high prevalence of visceral disorders and the widespread yet contested use of visceral techniques in osteopathy, a deeper understanding of current clinical practices is warranted. This is particularly relevant in Italy, where osteopathy was officially recognised as a healthcare profession in 2021 through national legislation enacted by presidential decree. According to pre-legislation data from the OPERA study (collected in 2017–2018), the estimated osteopath-to-population ratio in Italy was 8.0 osteopaths per 100,000 inhabitants, with approximately 5100 practitioners mainly working as self-employed professionals in private practice [16]. Additionally, national surveys suggest that healthcare professionals—including nurses, physicians, and physiotherapists—are increasingly recognising osteopathy as a valuable complementary discipline within multidisciplinary hospital settings, thereby supporting its integration into clinical practice [17,18]. Although visceral approaches are frequently employed, the discrepancy between clinical application and scientific evidence underscores the need to investigate the beliefs and clinical reasoning underlying their use [14,15].

The objective of this study is to explore the beliefs, reasoning processes, and experiential knowledge of expert Italian osteopaths regarding the management of visceral disorders.

## 2. Materials and Methods

### 2.1. Study Design

This qualitative study was conducted in accordance with the Standards for Reporting Qualitative Research (SRQR) [19] and the Consolidated Criteria for Reporting Qualitative Research (COREQ) [20] (see Appendix A), ensuring methodological transparency and rigour. A Constructivist Grounded Theory (CGT) approach was adopted to explore how experienced Italian osteopaths conceptualise and manage visceral disorders. Rooted in an interpretivist epistemology, CGT facilitates the co-construction of meaning between researchers and participants and is particularly suited to examining complex clinical reasoning and professional practices within their sociocultural context.

### 2.2. Research Team and Reflexivity

The research team comprised five osteopaths, TC (male), AB (male), FL (female), NR (male), and JEE (male). All researchers held BSc.Ost or MSc.Ost degrees; JE held additional academic qualifications, including a PhD. All authors were practising osteopaths; JE was additionally employed as a senior academic at ICOM Malta. JE, a senior academic with extensive expertise in qualitative methodology, supervised the project. TC, who conducted all interviews, had no prior personal or professional relationships with participants. To enhance reflexivity and minimise potential bias, the team maintained reflexive journals and engaged in regular debriefing sessions throughout the study. Team members meticulously documented reflections on potential interviewer influences during data collection, interpretation, and coding processes. These reflections focused on how the interviewer’s phrasing, assumptions, or clinical perspectives might shape participant responses. For instance, one team member noted: “The interviewer’s wording may have suggested a specific approach to visceral techniques evaluation, potentially influencing the participant’s reply.” Another observation highlighted: “A question was reformulated using the participant’s terminology, which might have subtly reinforced specific language use.” These critical reflections were systematically analyzed during debriefing sessions, serving two primary purposes: (1) to refine questioning techniques for subsequent interviews and (2) to inform analytic decisions. This reflective process was instrumental in maintaining the centrality of participants’ perspectives throughout the analysis, thereby enhancing the study’s validity and reliability.

### 2.3. Participants

Participants were selected using purposive sampling based on recognised expertise in visceral osteopathy. Recruitment was conducted via open invitations sent to publicly available email addresses listed in the national ROI (Registro degli Osteopati d’Italia) online website. The inclusion criteria required a minimum of 10 years of clinical experience, possession of a DO, BSc.Ost, or MSc.Ost qualification, and formal training in visceral osteopathy acquired either during undergraduate or postgraduate education. All participants were Italian osteopaths with no prior relationship with members of the research team. Demographic data—such as age, gender, years of clinical practice, and educational background—were collected and are summarised in Table 1. One eligible participant initially agreed to participate but was lost to follow-up before the interview could take place due to scheduling issues. Participants were informed that the study aimed to explore osteopathic approaches to visceral disorders and that the interviewer was also an osteopath with no personal or professional connection to them.

### 2.4. Setting

Interviews were conducted remotely via Google Meet in locations that were selected by participants to ensure comfort and confidentiality. One interview was conducted by telephone at the participant’s request. Each session lasted between 45 and 75 min. While only the interviewer interacted with the participant, one or more members of the research team were silently connected (with video and audio disabled) to monitor adherence to the interview guide and ensure methodological rigor. No repeat interviews were conducted. Each participant was interviewed once.

### 2.5. Data Collection

Data were collected through semi-structured interviews guided by a pilot-tested interview protocol (Table 2) developed based on relevant literature [20,21] and expert input. The interview guide covered four main domains: (1) training and continuing education in visceral osteopathy; (2) the definitions and identification of visceral disorders; (3) clinical management strategies and treatment approaches; and (4) interdisciplinary collaboration and patient education. The guide was iteratively refined to explore emergent themes [21]. All interviews were audio-recorded, transcribed verbatim, and anonymised. The interviews lasted between 29 min and 55 min, with an average duration of 36 min. To ensure data accuracy and enhance credibility, transcripts were returned to participants for verification and feedback (member checking) [17].

### 2.6. Data Analysis

Data analysis followed a two-stage coding process consistent with grounded theory methodology. In the initial coding phase, transcripts were segmented into discrete meaning units to identify preliminary concepts. This was followed by focused coding, in which related concepts were grouped into broader categories to support the development of theory. Each transcript was independently coded by at least two researchers, with discrepancies resolved through discussion and consensus. Reflexive memos were maintained throughout the process to document analytical decisions and support transparency. No qualitative analysis software was used; coding and data management were conducted manually by the research team. Theoretical saturation was considered achieved when, during the final interviews, no new categories, conceptual variations, or relevant insights emerged, and existing themes were consistently confirmed across participants. Saturation was assessed through constant comparison, reflexive memo writing, and iterative team discussions, following the standards recommended for qualitative research reporting [19]. The decision to conclude sampling was based on the repeated confirmation of existing categories and the absence of novel data from the last interviews. The diversity of participants’ educational and clinical backgrounds contributed to exploring a range of perspectives sufficient to support theory development within the scope of this study.

### 2.7. Ethical Considerations

The study received ethical approval from the Malta ICOM Educational Research and Ethics Committee (protocol code #2/2024; date of approval: 5 March 2024). Written informed consent was obtained from all participants before data collection. Data were anonymised and encrypted to ensure confidentiality. Participants were informed of their right to withdraw from the study at any time without consequence.

### 2.8. Trustworthiness

Multiple strategies were employed to enhance the trustworthiness of the study. Credibility was established through investigator triangulation, member checking, and peer debriefing. Dependability was supported by maintaining an audit trail and detailed documentation of the analytic process. Confirmability was strengthened through reflexive journaling and an external audit by an independent qualitative expert. Transferability was addressed by providing robust, contextualised descriptions of participant backgrounds and interview settings.

### 2.9. Data Sources

The literature review supporting the development of the study was conducted using PubMed, ScienceDirect, and Google Scholar. Search terms included “visceral disorder”, “visceral osteopathy”, and “visceral disorder and OMT”. Only peer-reviewed articles published in English were included. All key references informing the interview protocol and the interpretation of findings are cited in the manuscript. The datasets generated and analysed during the current study (i.e., anonymised interview transcripts) are not publicly available due to access restrictions imposed by the institutional Ethics Review Board and the conditions of the informed consent process. However, the data may be shared by the corresponding author upon motivated academic request, provided that confidentiality is maintained and the request complies with the study’s ethical framework. Novel methods are described in detail, while standard procedures are appropriately cited.

## 3. Results

Ten Italian osteopaths (P1–P10) participated in the study. Participants included both male and female professionals (age range: 40–68) with more than 10 years of clinical experience. All were actively practising in Italy at the time of the study. Most held a Diploma in Osteopathy (DO), often combined with undergraduate or postgraduate degrees in related disciplines such as Physiotherapy, Exercise Science, Medicine, and Chiropractic; one participant also reported holding a PhD. Every participant had experience in managing visceral disorders within the osteopathic context. Data analysis yielded four thematic categories: (a) training, experience, and continuing education; (b) the definition and identification of visceral disorders; (c) management and manual osteopathic treatment; and (d) multidisciplinarity, as can be seen in Table 3.

### 3.1. Training, Experience, and Continuing Education

Participants reported varied engagement with CPD, including literature review through databases (e.g., PubMed, ScienceDirect), and participation in postgraduate courses and conferences. Nonetheless, some admitted limited recent updating. In terms of clinical practice, several practitioners adhered to the Barral method, while others expressed skepticism regarding the anatomical specificity and efficacy of visceral palpation. Clinical experience was widely valued as essential in refining diagnostic sensitivity, particularly in complex cases. Peer exchange was perceived inconsistently—some saw it as a source of scientific enrichment, while others viewed it as redundant.

P6: “I’m unsure if my techniques are outdated... I probably wouldn’t be aware.”

P9: “Over time, I questioned: ‘How can I be truly certain?’”

P1: “Being absolutely sure that we are palpating the stomach rather than the intestine is not straightforward.”

### 3.2. Definition and Identification of Visceral Disorders

All participants agreed on the need to exclude underlying pathology through medical diagnosis. However, definitions of visceral disorders varied considerably. While some described them as functional alterations of visceral structures assessable through palpation or patient history, others provided experience-based, loosely defined interpretations. Diagnostic strategies diverged: some relied heavily on palpation to detect organ mobility or motility restrictions—even in asymptomatic patients—while others favored clinical history as the primary diagnostic tool. Regardless of approach, all acknowledged interrelations between visceral and musculoskeletal systems, often articulated using the metaphor of a “container-content” dynamic.

P3: “I trust much more what I feel than what the patient says... palpation gives me an idea of organ function.”

P7: “I identify visceral disorders primarily through manual evaluation.”

### 3.3. Management and Manual Osteopathic Treatment

Osteopathic management strategies for visceral disorders were highly heterogeneous. While some practitioners emphasized organ-specific manipulations aimed at restoring presumed dysfunctions, others prioritized systemic approaches, targeting the musculoskeletal system and autonomic regulation. Case history-taking practices varied: some used it as an entry point to identify contraindications and guide treatment planning, while others integrated it into ongoing manual assessment. Patient education was considered fundamental and ranged from anatomical explanations to lifestyle guidance. Communication strategies also differed—some deliberately avoided labelling findings as “problems” to minimize nocebo effects, while others openly discussed visceral palpation results. Criteria for monitoring outcomes ranged from subjective symptom reports to manual reassessment.

P3: “Palpatory skill is the ability to distinguish normal from paraphysiological...my hands indicate the dysfunctional area and its primary nature.”

P1: “Patient education... reporting carefully, setting measurable goals.”

P2: “I never say ‘you have a liver problem’... there might be no clinical issue.”

P4: “The insula regulates peristalsis and pain... I manipulate it first, then move to the intestine.”

### 3.4. Multidisciplinarity

Most participants endorsed referral to other healthcare professionals in cases of diagnostic uncertainty or when the clinical picture exceeded the scope of osteopathic practice. Multidisciplinary collaboration was generally valued, though practices differed. Some reported difficulties in communicating palpatory findings to peers or physicians, often due to terminology gaps or conceptual disagreements. One participant explicitly preferred working independently. Others emphasized the need to modernize osteopathic terminology to foster more effective interdisciplinary dialogue.

P1: “Though I’m an osteopath, I can’t say to a gastroenterologist that I manipulate the stomach... the terminology must evolve.”

P5: “I work within a multidisciplinary team, and I believe this approach is optimal.”

P6: “I refer patients to dietitians or nutritionists who know more than me.”

P3: “If specialized pathways haven’t begun, I refer—and always communicate briefly in writing.”

## 4. Discussion

In the academic year 2024–2025, two Italian universities (Verona and Florence) introduced university-level osteopathy programs, marking a significant shift in the professional landscape. These programs define a professional profile focused on preventing and treating musculoskeletal somatic dysfunctions, explicitly excluding medical pathologies [22]. The Italian osteopathic education system differs significantly from those in countries with established regulatory frameworks. While most Italian practitioners hold a Diploma in Osteopathy (DO), a smaller proportion have academic degrees from other European institutions [16]. This contrasts with countries like the United Kingdom or Australia, where standardized university-level education is the norm.

The fragmented educational landscape, mirrored in other national contexts [23], presents significant challenges. It impedes the consistent integration of evidence-based practice (EBP) and perpetuates tradition-based approaches. This fragmentation affects not only initial training but also continuing professional development, with a notable absence of a unified, evidence-based continuing education system [11]. While Italian osteopaths express interest in EBP, the lack of structured frameworks hinders its effective implementation [24]. This gap between interest and practice underscores the need for systematic changes in education and professional development.

These findings underscore the urgent need for structured, evidence-based undergraduate and professional education complemented by standardized continuing professional development programs. Such reforms are crucial for addressing the current professional fragmentation and facilitating a transition toward consistent, evidence-informed clinical practice. Given the complexity and prevalence of visceral disorders, a paradigm shift is necessary. The focus should shift from treating presumed visceral dysfunction to a more holistic approach that addresses individuals experiencing visceral-related symptoms. This approach often intersects with broader issues of chronic pain and persistent physical symptoms. Such reconceptualization aligns with contemporary healthcare models that emphasize person-centered care, therapeutic alliance, and the role of embodied cognition in shaping symptom experience and treatment response [25,26,27,28].

This transition towards evidence-based, person-centered osteopathic care represents a crucial step in the profession’s evolution. It not only enhances the credibility and effectiveness of osteopathic practice but also aligns it more closely with contemporary healthcare standards and patient expectations.

### 4.1. Interpretive Analysis of Definition and Identification of Visceral Disorders

Participants often exhibited inconsistent and vague definitions of visceral disorders, frequently relying on experiential knowledge. A common distinction was made between organic pathologies and functional disorders, with the latter being assessable using the Rome IV criteria [3,29]. Clinical evaluation typically begins with anamnesis and the exclusion of red flags, followed by manual assessment [30]. However, many participants prioritized palpatory findings over patient-reported symptoms despite the crucial role of the latter in EBP and patient-centered care [26].

Palpatory reliability remains low across studies, showing poor intra- and inter-examiner consistency regardless of the practitioner’s level of clinical experience. Factors influencing this include operator subjectivity, anatomical variability, and patient positioning [31,32,33,34,35,36,37]. The continued use of Barral’s palpatory-driven concepts of visceral mobility and motility, attributed to biomechanical, psychological, or traumatic origins, lacks empirical support and diagnostic reliability, and is prone to epistemological and cognitive bias [14,15,38,39,40,41].

The absence of shared definitions and the lack of critical reflection on the reliability of diagnostic processes render visceral osteopathy a self-referential, tradition-driven practice, where subjective perceptions supplant evidence and clinical reasoning. As highlighted in recent critiques of the profession, this diagnostic ambiguity risks perpetuating implausible models and biologically inconsistent concepts, thus limiting osteopathy’s scientific and clinical credibility [10,42].

This epistemological fragility translates into marked diagnostic variability across practitioners: while some rely exclusively on palpatory findings, others combine them with patient-reported symptoms and case history. These divergent approaches do not reflect conscious methodological choices but rather the legacy of fragmented, tradition-based training, which shapes clinical reasoning in the absence of shared diagnostic models.

A robust, patient-centered framework integrating clinical experience with empirical data and patient values is essential for progress in this area [43]. Although the therapeutic alliance is increasingly recognized as a central component of effective clinical practice and osteopathic care, patient–practitioner interactions can be more appropriately understood through broader frameworks. Embodied cognition offers valuable insights into this context, emphasizing the interconnectedness of mind, body, and environment in shaping cognitive processes and clinical interactions. This approach recognizes that clinical reasoning and decision-making are not purely intellectual exercises but are deeply influenced by bodily experiences, sensorimotor processes, and environmental factors. In osteopathic practice, embodied cognition can inform how practitioners perceive, interpret, and respond to patients’ physical and verbal cues, potentially enhancing diagnostic accuracy and treatment effectiveness. Moreover, it can provide a more nuanced understanding of the patient’s lived experience of their condition, fostering a more empathetic and holistic approach to care. Integrating principles of embodied cognition into osteopathic education and practice could lead to more refined, context-sensitive clinical skills and a deeper understanding of the complex, dynamic nature of the therapeutic encounter. Such enactive and ecological models of osteopathic care provide a more robust conceptual and methodological foundation for learning and improving patient–practitioner interactions [25,28].

### 4.2. Osteopathic Management and Treatment

Participants widely reported using organ-specific techniques based on visceral mobility and motility, despite the rationale being self-referential and evidence for their clinical efficacy remaining limited [2,14,15]. The use of craniosacral techniques to influence visceral function remains speculative and unsupported by scientific evidence [44,45]. Claims of modulating visceral activity through manual stimulation of brain regions raises substantial conceptual concerns and risks of perpetuating biologically implausible models [42]. While osteopathic interventions may influence autonomic regulation, the mechanisms are likely multifactorial and involve contextual and relational factors [46,47].

Patient education was emphasized by participants, including recommendations on lifestyle factors such as diet, hydration, sleep, physical activity, and posture [48,49,50,51,52,53]. However, basing such advice solely on palpatory findings may lack clinical validity [2]. Terminology also plays a crucial role in shaping patient perception, as outdated or ambiguous phrases may induce nocebo effects and contribute to confusion. In this regard, the continued use of osteopathic jargon such as “visceral dysfunction” can be particularly problematic, as it implicitly suggests the presence of a non-clearly detectable pathological state, potentially misleading both patients and other healthcare providers [9,54].

Therapeutic progress was often monitored solely through palpation, with limited use of standardized outcome measures. The integration of patient-reported outcome measures (PROMs) and validated scales is recommended to align with person-centered care and enhance data reliability [27]. While no single validated questionnaire captures every facet of clinical evaluation, various Patient-Reported Outcome Measures (PROMs) have already been developed to address different aspects of gastrointestinal symptoms and related experiences. Examples include the Visceral Sensitivity Index (VSI), the Gastrointestinal Symptom Rating Scale (GSRS) and Quality of Life in Reflux and Dyspepsia (QOLRAD) scale, and the Leuven Postprandial Distress Scale (LPDS). These tools illustrate how standardized instruments can offer insights into anxiety, symptom burden, and quality of life, providing potential reference points for more consistent and multidisciplinary approaches [55,56,57].

Using these PROMs may improve assessment consistency and support integration into multidisciplinary care. This approach could enhance the assessment process by providing standardized, quantifiable data to complement manual evaluations. Additionally, incorporating decision-support tools based on these validated measures could aid in creating more objective and reproducible diagnostic criteria.

It is important to note that, while these tools can provide valuable insights, the osteopathic decision-making process should still prioritize the individual patient’s symptom presentation. The integration of PROMs and other validated tools should serve to support and enhance clinical judgment rather than replace it entirely.

This hybrid model, combining patient-reported data with manual assessment, could potentially bridge the gap between traditional osteopathic approaches and evidence-based practices in multidisciplinary settings. It may also facilitate better communication and collaboration among healthcare providers from different specialties.

The heterogeneity in treatment strategies for visceral-related problems in osteopathic care, ranging from organ-specific interventions to systemic and patient-centered approaches, reflects inconsistent diagnostic models rather than deliberate clinical reasoning. This variability stems from fragmented and tradition-based education, creating a cycle where diagnostic inconsistencies directly influence treatment decisions. The interconnected nature of education, diagnostic reasoning, and treatment strategies perpetuates outdated and inconsistent clinical practices. This cycle persists due to a lack of critical reflection and evidence-informed frameworks in osteopathic care for visceral-related problems.

### 4.3. Screening for Referral, Multidisciplinarity, and Communication with Other Healthcare Professionals

All participants endorsed referral in the presence of red flags or diagnostic uncertainty [30]. Interdisciplinary collaboration was generally viewed positively, particularly for multifactorial conditions such as DGBI [58]. Nonetheless, inconsistencies in language and conceptual framing often hindered effective communication with other health professionals.

Descriptive expressions based solely on palpation (e.g., “heavy liver”, “struggling kidney”) lack diagnostic specificity and, although they are elements of traditional osteopathic practice, should not singularly guide clinical reasoning and may compromise interprofessional dialogue. Aligning osteopathic terminology with contemporary biomedical standards is essential for fostering shared decision-making and cohesive care strategies [59].

To advance multidisciplinary integration, Italian osteopaths must adopt standardized, transparent communication practices that facilitate collaboration. Doing so will improve care continuity, reduce fragmentation, and enhance patient engagement in treatment [25,58,60].

The provided table (Table 4) effectively addresses the reviewers’ comments by offering a clear comparison between outdated terminology and recommended alternatives in visceral osteopathy. This table serves as a valuable guide for osteopaths to adopt more standardized and communicable language.

Adopting this updated terminology in clinical practice and research is crucial for improving communication, reducing misunderstandings, and promoting evidence-based approaches in visceral manipulation and osteopathy. By using more neutral and descriptive language, practitioners can enhance patient understanding, foster collaboration with other healthcare professionals, and align their practice with current scientific standards.

### 4.4. Limitations

This study has several limitations. First, the interviews and thematic analysis were conducted by researchers with professional experience in osteopathy, which, despite the use of reflexive journals and peer debriefing, may have introduced interpretive bias. Second, the sample size was small (*n* = 10), and although qualitative research does not aim for statistical generalizability, caution should be exercised when extrapolating these findings to the broader Italian osteopathic community.

An additional limitation concerns the heterogeneous background of the participants, some of whom also practised as medical doctors, physiotherapists, or kinesiologists. This interdisciplinary profile, while reflective of the reality of osteopathic practice in Italy, may have influenced responses and limited the applicability of findings to osteopaths with more homogeneous training. Furthermore, participant recruitment was based on voluntary response to open invitations without stratified selection criteria. This may have introduced a self-selection bias, favoring individuals more engaged with professional development or reflective clinical practice. Moreover, although the interview guide was consistent across participants, variability and clarification in question phrasing, potentially carried by limited prior experience in qualitative interviewing, were used to explore emerging concepts and may have influenced participant responses. Finally, the absence of observational or triangulated clinical data restricts the extent to which reported practices can be objectively validated.

## 5. Conclusions

This qualitative study examined the epistemological, clinical, and communicative aspects of the osteopathic management of patients with visceral disorders in Italy through in-depth interviews with experienced practitioners. Our findings reveal significant heterogeneity in educational backgrounds, theoretical models, and therapeutic approaches among osteopaths, often influenced more by tradition and subjective interpretation than by scientific evidence. The study identified several key issues: fragmented education and a lack of standardised curricula, overreliance on palpatory assessment with limited diagnostic reliability, insufficient emphasis on patient-reported outcomes, persistence of tradition-based models in the absence of evidence-based frameworks, and challenges in interprofessional communication and integration within healthcare systems.

Based on these findings, we propose a grounded theory framing visceral osteopathic practice in Italy as a clinical–cultural continuum. This continuum, shaped by educational fragmentation, epistemological ambiguity, and adherence to traditional models, has a significant influence on diagnostic reasoning, treatment choices, and professional communication. Our findings align with challenges observed in other European contexts, reflecting broader continental trends in osteopathic professionalization. However, this study is limited by its focus on Italian practitioners and may not fully represent the diversity of osteopathic practice across Europe.

To address these challenges, we recommend developing standardised, evidence-based curricula for osteopathic education, promoting structured continuing education programs, implementing validated clinical assessment tools, encouraging critical engagement with emerging research, adopting updated, standardised terminology to enhance interprofessional communication, and fostering collaboration with other healthcare professionals. These steps are crucial for enhancing the legitimacy of osteopathy in Italy and facilitating its integration into the broader healthcare system.

Future research should involve larger and more diverse samples, ideally triangulated with observational data, to test the transferability of these findings across broader osteopathic contexts. Additionally, studies should focus on evaluating the effectiveness of these recommendations and exploring their applicability in diverse healthcare contexts. This study contributes to the ongoing dialogue about the evolution of osteopathic practice in Europe.

## Figures and Tables

**Table 1 healthcare-13-01995-t001:** Demographic and educational characteristics of the participants.

Px	Sex	Age	Prior Education	Years of Experience
P1	Female	42	BSc.Ost, MSc.Ost.	19
P2	Male	44	BSc.Sports and Exercise Science, part-time DO	13
P3	Female	48	BSc.Physiotherapy, DO	21
P4	Male	65	BSc.Physiotherapy, DO	24
P5	Male	68	BSc.Ost., BSc.Physiotherapy, Doctor in Chiropractic, PhD	40
P6	Male	67	BSc.Sports and Exercise Science, DO Doctor of Medicine	24
P7	Female	40	DO	10
P8	Male	47	BSc.Physiotherapy, part-time DO, Doctor of Medicine	18
P9	Female	44	BSc.Sports and Exercise Science, part-time DO	12
P10	Male	60	BSc.Physiotherapy, part-time DO	26

**Table 2 healthcare-13-01995-t002:** Overview of the semi-structured interview guide. The table outlines the thematic domains and corresponding questions used to explore participants’ beliefs and clinical reasoning in the osteopathic management of visceral disorders.

Training:	“What has been your training in the field of osteopathy? And specifically in visceral osteopathy?”
Professional development	“How has your practice evolved over the years?”
Definition of visceral disorder:	“In the clinical context, how do you identify a visceral disorder in your osteopathic practice?” “How would you define a visceral disorder?”
Practical application:	“How does your clinical reasoning translate into manual treatment for a patient with a visceral disorder?”
Management:	“How would you define the ‘management’ of a patient with a visceral disorder?” “What elements do you assess in the management of such patients?” “How do you interpret the concept of management?”
Integration with other osteopathic contexts:	“How do you integrate the management of visceral disorders within the broader framework of your osteopathic practice?”
Outcome evaluation:	“How do you monitor the effects and progress of your treatments in patients with visceral disorders?”
Multidisciplinarity:	“Do you collaborate with other healthcare professionals in managing patients with visceral disorders? If so, how?”
Patient education:	“Do you provide education to patients regarding the management of their visceral disorders? If so, how?”
Relevant clinical experience:	“Is there a clinical case you consider significant in the context of managing a patient with a visceral disorder?”
Continuing education:	“How do you stay up to date regarding the management of patients with visceral disorders?”

**Table 3 healthcare-13-01995-t003:** Summary of thematic categories and representative findings.

Category	Main Themes	Representative Quotes
Training, Experience, and Continuing Education	Variable engagement with CPD; frequent reliance on Barral method; mixed attitudes toward the reliability and relevance of visceral palpation and its related techniques.	P5: “The method I use is Barral”
Definition and Identification of Visceral Disorders	Inconsistent definitions of visceral disorders across practitioners; varying reliance on palpatory findings versus patient history for diagnosis; widespread acknowledgement of viscero-somatic interactions.	P3: “I trust much more what I feel than what the patient says...”
Management and Manual Osteopathic Treatment	Heterogeneous treatment strategies: some practitioners focused on organ-specific techniques aimed at correcting perceived dysfunctions, while others prioritized systemic approaches targeting autonomic regulation; emphasis on patient education associated with variable communication strategies: some practitioners deliberately avoiding pathological labels to reduce nocebo effects, while others openly discussing palpatory findings.	P2: “I never tell a patient they have a visceral issue...”
Multidisciplinarity	Generally positive attitude toward multidisciplinary collaboration, with referral to other healthcare professionals in case of diagnostic uncertainty; however, limited interdisciplinary communication persists due to terminology inconsistencies and conceptual divergences; some practitioners work in isolation.	P1: “I always refer when there’s no diagnosis...”

**Table 4 healthcare-13-01995-t004:** Examples of legacy terms commonly used in visceral osteopathy and suggested alternatives to improve clinician–patient communication and interprofessional dialogue.

Legacy Term, Commonly Used	Problematic Aspects	Possible Alternatives	Why It Matters
Visceral dysfunction	Vague, implies a pathological state not better defined; confusing for patients and colleagues	Visceral-related complaint/visceral-related symptoms	Neutral and descriptive; avoids implying hidden disease
Organ restriction	Suggests a structural blockage, which cannot be objectively verified	Perceived limitation in tissue mobility and motility	Clarifies that it is a clinician’s perception, not an anatomical fact
Blockage	Alarmist language; could raise unnecessary concerns	Area of perceived tension	Softens the term, avoids suggesting obstruction
Correction of dysfunction	Implies a deterministic “fixing” of the organ	Supportive manual approach	Emphasises patient-centered care and avoids overclaiming effects

## Data Availability

The data presented in this study are available from the corresponding author upon reasonable request, owing to access restrictions imposed by the institutional Ethics Review Board and the conditions of the informed consent process.

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
