# Peer review of "Clinical Reasoning and Practices in the Osteopathic Management of Visceral Disorders: A Grounded Theory Study in the Italian Context"

_healthcare, 2025, doi:10.3390/healthcare13161995_

Round 1
Reviewer 1 Report
Comments and Suggestions for Authors
- Two periods on the line 109
- I have a doubt, Could the participants read the interview transcriptions?
Author Response
Comment 1: Two periods on the line 109.
Response 1: Thank you for noticing and pointing this out. We have corrected the duplicated period in line 109.
Comment 2: I have a doubt, Could the participants read the interview transcriptions?
Response 2: Thank you for your question. As clarified in the Methods section, transcripts were returned to participants for verification as part of the member checking process. This is explicitly stated in the sentence: “To ensure data accuracy and enhance credibility, transcripts were returned to participants for verification (member checking) [17].”
We thank all reviewers for their constructive feedback. We provide a detailed point-by-point response, indicating the changes made in the revised manuscript.

Reviewer 2 Report
Comments and Suggestions for Authors
Dear Authors,
The manuscript is well written. In complex not required relevant improvements. Title and Abstract ok. In the keywords part, I suggest the use max. 5 words with the relevant details as type of study and setting. In the introduction could be interest to ecxtend the perception of healthcare wokers of osteopathy in the context of the study. There are relevants studies in recent time (2024-2025) that can complete the section and amply the intererest of data finding. The tool declared as tool missing of check list in supplementary material. The discussion are well structured: the seggest poned in the introduction could be adopted in this section too. Native review suggested.
Comments on the Quality of English LanguageNative review suggested
Author Response
We thank all reviewers for their constructive feedback. We provide a detailed point-by-point response, indicating the changes made in the revised manuscript.

Reviewer 3 Report
Comments and Suggestions for Authors
I want to thank the editor for the opportunity of reviewing this qualitative study. The manuscript is well written and the English grammar is good under in my humble opinion as non native speaker. The presentation is adequate with all the needed section for this type of study. Please consider some issues:
Abstract: the objective of the study is missing
Line 46-47: a cite supporting this affirmation is needed.
Methods section: investigation has been conducted in a proper way, with a good interview design and adequate participants selection. No changes are suggested. Results and Discussion sections are properly conducted.
All possible biases have been addressed in limitations section. I congratulate the authors for highlight their role as osteopaths as possible bias.
Author Response

(The authors gave the same response as above.)

Reviewer 4 Report
Comments and Suggestions for Authors
Dear authors
Please find attached

Author Response

(The authors gave the same response as above.)

Reviewer 5 Report
Comments and Suggestions for Authors
The article, which explores the clinical reasoning and practices adopted by Italian osteopaths in the management of visceral disorders, using a qualitative constructivist grounded theory approach. The topic is relevant and addresses an area with a known gap in empirical literature, particularly in the context of evolving professional regulation in Italy. The aim to map clinical beliefs and practices in the controversial field of visceral osteopathy is valid and timely.
However, upon detailed review, we find that the manuscript, in its current form, presents significant limitations that reduce its scientific contribution. First, the empirical basis of the study rests on a small sample of ten participants, recruited through convenience sampling, with limited description of geographic or institutional diversity. Although common in qualitative designs, this limitation is not sufficiently acknowledged, nor are robust strategies described to ensure theoretical saturation or external validity of findings.
Second, while the data analysis is structured around thematic categories, it leans heavily on descriptive quotes and participant accounts, with limited theoretical abstraction. For a study aiming to generate grounded theory, one would expect a stronger interpretative effort—identifying explanatory mechanisms, mapping variability across cases, and articulating interrelations among categories. As it stands, the analysis remains largely descriptive and falls short of producing theory with transferability beyond the sample.
Another major concern is the epistemological stance of the authors. Although the manuscript rightly critiques the limited scientific evidence supporting visceral osteopathy—particularly the low diagnostic reliability of palpation and the questionable validity of certain techniques—it nonetheless presents these same practices in a largely neutral, sometimes uncritical manner. This ambiguity undermines the scientific positioning of the article and risks inadvertently normalizing practices that are not evidence-based.
The study would benefit greatly from a stronger alignment with contemporary models of clinical reasoning, such as embodied cognition, person-centered care, or enactive-ecological frameworks. While some of these are mentioned in the discussion, they are not used as guiding structures in the analysis. In addition, a deeper reflection on the role of educational systems, cultural factors, and regulatory gaps in perpetuating outdated practices would enhance the depth and utility of the manuscript. Clearer and more constructive recommendations for curricular reform and professional development would also be welcome.
Finally, although the manuscript is well written and methodologically transparent, its contribution to the international literature remains modest if it does not move beyond descriptive mapping of beliefs and practices that have already been documented in grey literature and previous reports. We therefore recommend substantial revision, with more robust theoretical engagement, explicit critical positioning, and a stronger focus on the implications for evidence-based practice.
Author Response

(The authors gave the same response as above.)

Round 2
Reviewer 2 Report
Comments and Suggestions for Authors
Dear Authors,
very good job. Ready for publication.
Best
Author Response
Thank you for your kind response
Best regards
Reviewer 5 Report
Comments and Suggestions for Authors
I have carefully examined the revised manuscript titled "Clinical Reasoning and Practices in the Osteopathic Management of Visceral Disorders: A Grounded Theory Study in the Italian Context". The authors have made commendable progress in addressing the concerns raised during the initial round of peer review. The manuscript now demonstrates improved methodological clarity, stronger theoretical grounding, and a more structured discussion of the results. Notably, the integration of embodied cognition and enactive care models has enriched the epistemological foundation of the work and aligned the discussion with contemporary person-centred healthcare paradigms.
The revised methodology section benefits from greater transparency, particularly in the description of sampling, interview procedures, coding processes, and reflexive practices. The segmentation of the findings into thematic categories is coherent and well-supported by illustrative quotations from participants. Furthermore, the discussion critically examines diagnostic variability, palpatory reliability, and the implications of fragmented educational pathways in Italian osteopathy. These insights make a valuable contribution to the understanding of professional identity and practice variation within the field.
However, several points require further refinement before final acceptance. First, while the manuscript acknowledges the limitations of traditional osteopathic terminology, such as "visceral dysfunction", it would benefit from a more explicit elaboration of how these terms may impact clinical relationships and interprofessional communication. The authors might consider including a table comparing outdated versus recommended terminology to guide osteopaths towards more standardised and communicable language.
Second, although the discussion rightly critiques the reliance on palpatory findings, it would be useful to expand upon potential alternative assessment methods that may enhance diagnostic consistency and integration into multidisciplinary settings. This could include reference to validated outcome measures, decision-support tools, or hybrid models combining patient-reported data with manual assessment.
Third, while the reflexive stance of the research team is noted, it would strengthen the manuscript to provide more concrete examples of how reflexivity was operationalised—perhaps through brief excerpts from reflexive journals or team debriefings. Such transparency would demonstrate how the authors actively mitigated potential bias, given their professional proximity to the subject matter.
Fourth, although the sample size is appropriate for grounded theory research, the authors should reiterate in the conclusion that future studies involving larger and more diverse samples—or triangulated with observational data—are warranted to test the transferability of findings across broader osteopathic contexts.
Author Response
Reviewer 5
Comment 1: First, while the manuscript acknowledges the limitations of traditional osteopathic terminology, such as "visceral dysfunction", it would benefit from a more explicit elaboration of how these terms may impact clinical relationships and interprofessional communication. The authors might consider including a table comparing outdated versus recommended terminology to guide osteopaths towards more standardised and communicable language.
Response 1: We have expanded the Discussion to further elaborate on the impact of legacy terminology on clinician–patient relationships and interprofessional communication. In addition, we created a new table (Table 4) that compares commonly used osteopathic terms (e.g., visceral dysfunction) with proposed alternatives aligned with neutral and evidence‑informed
Comment 2: Second, although the discussion rightly critiques the reliance on palpatory findings, it would be useful to expand upon potential alternative assessment methods that may enhance diagnostic consistency and integration into multidisciplinary settings. This could include reference to validated outcome measures, decision-support tools, or hybrid models combining patient-reported data with manual assessment.
Response 2: We have expanded Section 4.3 to include alternative assessment strategies, referencing validated PROMs (e.g., VSI, LPDS, QOLRAD, GSRS) and outlining how they can be integrated into osteopathic care.
Comment 3: Third, while the reflexive stance of the research team is noted, it would strengthen the manuscript to provide more concrete examples of how reflexivity was operationalised—perhaps through brief excerpts from reflexive journals or team debriefings. Such transparency would demonstrate how the authors actively mitigated potential bias, given their professional proximity to the subject matter.
Response 3: We have revised Section 2.2 (Research Team and Reflexivity) to include a concrete example of how reflexivity was operationalised during the study. Specifically, we added sentences illustrating how team members noted potential influences of interviewer phrasing on participants’ responses
Comment 4: Fourth, although the sample size is appropriate for grounded theory research, the authors should reiterate in the conclusion that future studies involving larger and more diverse samples—or triangulated with observational data—are warranted to test the transferability of findings across broader osteopathic contexts.
Response 4: We have revised the conclusion to explicitly address the need for larger and diverse samples
